# WGS Data Collections: How Do Genomic Databases Transform Medicine?

**DOI:** 10.3390/ijms24033031

**Published:** 2023-02-03

**Authors:** Zbigniew J. Król, Paula Dobosz, Antonina Ślubowska, Magdalena Mroczek

**Affiliations:** 1Central Clinical Hospital of Ministry of the Interior and Administration in Warsaw, 02-507 Warsaw, Poland; 2Department of Biostatistics and Research Methodology, Faculty of Medicine, Collegium Medicum, Cardinal Stefan Wyszyński University in Warsaw, Wóycickiego 1/3, 01-938 Warsaw, Poland; 3Center for Cardiovascular Genetics and Gene Diagnostics, Foundation for People with Rare Diseases, Wagistrasse 25, 8952 CH-Schlieren-Zurich, Switzerland

**Keywords:** COVID-19, database, genomic databases, whole-genome sequencing, WGS

## Abstract

As a scientific community we assumed that exome sequencing will elucidate the basis of most heritable diseases. However, it turned out it was not the case; therefore, attention has been increasingly focused on the non-coding sequences that encompass 98% of the genome and may play an important regulatory function. The first WGS-based datasets have already been released including underrepresented populations. Although many databases contain pooled data from several cohorts, recently the importance of local databases has been highlighted. Genomic databases are not only collecting data but may also contribute to better diagnostics and therapies. They may find applications in population studies, rare diseases, oncology, pharmacogenetics, and infectious and inflammatory diseases. Further data may be analysed with Al technologies and in the context of other omics data. To exemplify their utility, we put a highlight on the Polish genome database and its practical application.

## 1. Introduction

It has been over 20 years since the very first version of the entire human genome was released [1]. Today, although advanced sequencing methods are available at a reasonable price and the role of significant genetic variants localised along the whole genome is quite well defined, the clinical implementation of whole-genome sequencing (WGS) in diagnosis and treatment remains in its infancy [2,3,4]. Whole exome sequencing was thought to elucidate the genetic background of most of the inherited diseases. However, it was not the case, which is why other omics technologies, such as WGS, RNA-Seq, Epigenomics and Metabolomics, gained importance [5,6,7]. Here, we would like to highlight the importance of creating WGS databases. With an increasing number of individuals included in the databases, it became apparent that genetic variation differs significantly across ethnic groups [8]. Therefore, it is necessary to create local genomic databases that mirror the smaller and sometimes even endemic population structure.

Oncology remains the major field that can benefit the most from whole genome sequencing since cancer develops because of the changes in the DNA inside cells [9] and also given the number of individuals affected. Thus, we should not hesitate to say that cancer actually is a disease of our genome. Traditionally, most of the studies were focused on the identification of cancer mutations solely in protein-coding genes, ignoring the remaining 99% of the genome dubbed as “the junk DNA”, partially because very few tools for big data analysis were available, if any at all [10]. As a result, the first collection of more than 350 cancer-related genes, protein-coding genes to be more specific, has been created with new genes being added over time [11]. Currently used cancer diagnostic panels span more than 500 genes forming the modern foundation of cancer diagnosis [12]. However, gene panels identify mutations only in 0.01–0.10% of the genome, at best. The identification of variants scattered in the whole sequence of the genome may provide significantly more precision in cancer diagnosis and better treatment options.

The remaining 99% of the total DNA sequence, colloquially called “dark matter of the DNA” is much less characterised, but also holds disease-relevant changes [10]. Multiple types of RNA produced from those regions regulate gene expression at many levels and this knowledge dramatically changed our understanding of how disease arises and progresses [12,13]. Whole Genome Sequence (WGS)-based analyses of thousands of genomes representing various cancer tissues revealed multiple cancer-driver events localised in non-coding regions of DNA such as promoters, enhancers, or miRNA coding genes [14], to name just a few. Such events include not only single nucleotide variations (SNV) but also small indels and larger structural changes [14]. Although the driver mutations identified in non-coding regions are less frequent (13%) than in protein-coding genes (87%), these variant numbers will grow with more cancer genomes sequenced [14]. Moreover, in terms of exome region sequencing, WGS is more powerful than WES [15], also in terms of detecting structural variants and exome coverage [16]. Fine-tuning cell functioning using this whole new category of personalised therapies, or targeting specific targets in non-coding regions, might have tremendous results, arming us with new powerful tools in intervening and treating human diseases, cancer in particular. Big data analysis is a game-changer, and we can be certain that the remaining unknown chunk of the DNA is important, even if we are at the very beginning of the road to fully understand it.

The number of already existing genomic databases is increasing. The journal *Nucleic Acids Research* (NAR) publishes annually a special issue on Molecular Biology Databases, a considerable proportion of which is related to genomics. The number of NAR’s papers reporting new databases in the last ten years has reached nearly 700 (Figure 1) [17,18,19,20,21,22,23,24,25,26]. Most of the databases, except from a few exceptions, contain pooled data not only from genomes, but also from exomes, RNA-Seq or epigenomic data. Additionally, they differ in terms of informed consent given by the participant [27].

They find their application in various fields of medicine and their use in research generally increases over time. This phenomenon can be illustrated with a comparison of the number of results of the PubMed search query: “Name of database” in the last ten years for two databases described in the next part of the review dedicated to oncology (Figure 2). The first one is The Cancer Genome Atlas (TCGA). The second one, the Exome Aggregation Consortium (ExAC), transformed into the Genome Aggregation Database (gnomAD), is widely used in the population studies of all types, cancer studies included.

In this review, we present the examples of the existing and currently being developed genomic databases and their possible use. We also discuss challenges and possible limitations to these global efforts and future means of improvement, to be implemented not only in oncology or infectious diseases, but also other areas of medicine. We strongly believe that modern medicine cannot exist without genomics.

## 2. Genomic Databases: Global vs. Local Initiatives

Several efforts focused on creating large aggregating genome databases gathering data across different population of diverse ancestries. Some projects gathered a considerable number of samples, such as the TopMed Program with 53831 genome-sequenced samples [28] or gnomAD with 15,708 genomes in addition to exomes [29] (Figure 3). One of the biggest databases that is publicly accessible gathering both phenotypical and genetic databases is UK biobank, which released WGS data from nearly 200.000 individuals, mainly British, but also from Asian and African origin [30]. On the pan-European level a well-established initiative is 1000 Genomes Project (1kGP), which collected 3202 genomes from diverse ancestries, among them 602 trios [31]. A bigger project, the 1+ Million Genomes Initiative, is coordinating the efforts to provide a proper infrastructure and framework according to local regulations to work on the genomic data [32]. On the global level The National Center for Biotechnology Information (NCBI) curates a large genomic database, Human Genome Resources [33], encompassing small variations, structural genomic changes and information on the relation of genomic variation to human health. Additionally, investigating infectious diseases such as COVID required pooling of the large genetic data from many populations and across diverse ancestries. One such initiative is the COVID Human Genetic Effort (HGE), collecting the genomic data across the populations for the scope of identifying variants influencing the disease course [34,35,36].

The non-European genomic databases are still underrepresented in the global databases with pooled data and a study from 2009 showed that 96% of genetic data are from studies performed solely on Europeans [37]. However, several of the gaps have been covered by local projects (Figure 3). Recently, very ambitious efforts in terms of WGS databases have been undertaken in Africa and Asia. In Africa a large consortium plans to WGS sequence 3 million genomes [38]. The initiative will be a response to public health challenges, such as parasitic and infectious diseases, especially HIV. The Genome Asia 100K project plans to sequence 100,000 individuals from 12 South Asian, and at least 7 North and East Asian countries [39]. A genome sequencing initiative for Japan has also already been implemented [40]. In the Middle East, more than 7000 individuals have been genome-sequenced in recent years [41]. Importantly, it revealed that polygenic scores (PGS) have reduced predictive performance in the Qatari population, postulating the necessity of including population-specific data in the PGS studies.

Although some genomic data are aggregated as part of the biggest projects, still there is a need to create smaller, population-specific databases that can reveal a different genetic landscape on the local level than a pan-European project. Given the ancestry component in medical genetics, it is important that huge datasets reflect the populations of interest. One such genomic database considering the structure and background of the investigated subjects is PGG.Population [42]. The database gathered >7000 genomes from 356 global populations and is designed for population genetic studies. On the European level, several European countries started their own biobanks, including genomic data. The leaders in this field are Iceland [43], Finland [44] and Estonia [45]; however, the majority of European countries is curating its own genomic database, in addition to their contribution to global efforts [46]. Slavic genomes are often underrepresented [47,48]; therefore, a recent Polish initiative to create a database of 1076 unrelated Poles is an important step forward into including less represented European populations [48]. Several other initiatives for Slavic genomes have been undertaken, including in the Ukraine [47]. We have a personal experience on this project created initially to investigate the course of COVID disease. As the participants consented to the scientific use of their anonymised data, the database can be used as a reference database for the Polish population. The data of the participants that consented might be deposited as part of the Genome of Europe project.

Furthermore, some of the analyses, especially evolutionary events, justify the creation of smaller local databases. A Japanese genomic database revealed evolution traits related to alcohol or nutrition [49]. In Asia local projects sequencing local populations have arisen, such as KOVA 2, collecting almost 2000 genome-sequenced Korean individuals, surpassing, together with WES data collected in KOVA 2, the data on Korean individuals collected in gnomAD [50]. The whole genome sequencing of Ethiopian highlanders delineated genomic regions that may have an impact on hypoxia tolerance [51]. In Europe there is a tendency to create local genomic databases that are especially important in interpreting VUS and in the application of PGS on the local level [46].

## 3. Genomic Databases in Oncology

As previously mentioned, thanks to genomics, our knowledge of cancer biology has expanded considerably in the past few years. The rapid development of genomic research in this field would be impossible without the joint efforts of the scientific community to generate public databases, which have been extensively used as a tool for further studies on various aspects of oncology.

By the end of 2005, the US National Institutes of Health launched The Cancer Genome Atlas (TCGA) project. In 12 years, TCGA characterised more than 20,000 samples from 33 different cancer types, generating over 2.5 petabytes of genomic, epigenomic, transcriptomic and proteomic data [52]. As it soon turned out that characterising a higher number of tumour samples from different cancer types would require international cooperation, the International Cancer Genome Consortium (ICGC) was initiated in 2008 to coordinate large-scale cancer genome sequencing studies in 50 different tumour types “that are of clinical and societal importance across the globe” [53]. It is the most significant cancer genome sequencing project to date. Over 80 million somatic mutations have been identified in this dataset. Both TCGA and ICGC were mainly focused on the exome. However, several studies have shown the important role of non-coding and regulatory regions in carcinogenesis. That is why the Pan-Cancer Analysis of Whole Genomes (PCAWG) initiative within the ICGC was established to identify common patterns of mutation in more than 2600 cancer whole genomes. According to the flagship paper of the TCGA/ICGC PCAWG consortium published in 2020, the majority of cancer genomes contain a few driver mutations in both coding and non-coding regions but in about 5% of them, no known mutation was identified, which leaves room for speculation. Is the catalogue of cancerogenic mutations incomplete or do other processes have more impact in these cases? The new phase of the project Accelerating Research in Genomic Oncology (ARGO) started in 2019. Its main goal is to improve the outcome of cancer patients. It will analyse 100,000 samples in comparison with clinical data to find out how to best use genomic knowledge in the prevention, detection and treatment of cancers [52,54].

Projects, as described above, have enabled the creation of such data collections as the Catalogue of Somatic Mutations in Cancer (COSMIC), the world’s largest and most detailed resource for exploring the effect of somatic mutations in human cancer, and the Cancer Gene Census (CGC) [55,56]. COSMIC covers all the known genetic mechanisms by which somatic mutations promote cancer such as coding and non-coding mutations, gene fusions, copy-number variants, and drug-resistance mutations, whereas CGC is an expert-curated catalogue of the genes driving human cancer that is used as a standard in cancer genetics across basic research, medical reporting and pharmaceutical development. It also includes functional descriptions of how each gene contributes to disease generation [55,56].

Another large database widely used in oncological research, as well as in other domains, is the Genome Aggregation Database (gnomAD), originally launched in 2014 as the Exome Aggregation Consortium (ExAC) [57]. It contains over 125,000 exome and 15,000 whole genome sequences from European, Latino African and African American, South Asian, East Asian, Ashkenazi Jewish and other populations (https://gnomad.broadinstitute.org (accessed on 10 December 2022)). All the data were contributed to the project from independent large-scale human sequencing studies led by more than 100 investigators, then processed into summary high-quality variant data and made available for the wider scientific community. The gnomAD papers report 241 million small genetic variants and over 335 thousand structural variants [57]. Even though this database is widely used in oncology, it remains a valuable and broad population database with many significant applications outside medicine too.

In addition to the already listed, numerous smaller, more specific databases have been created. Some of the interesting examples include a database of extrachromosomal circular DNA (eccDNAdb), which seems to play a crucial role in oncogene amplification and tumour progression [58]; single nucleotide polymorphisms (SNPs) databases, (SNPs can influence methylation and participate in signalling pathway degeneracy in cancer) [59] and upstream open reading frames (uORFs) databases. Genetic defects in the last ones have been linked to the development of various diseases, including cancer [60].

All these resources are used in cancer-related analyses. They allow detection of viral sequences in cancer tissues, e.g., herpesvirus family or HPV in bladder cancers [61]; finding new genetic markers to diagnose and treat diseases with relatively poor prognosis such as liver and oesophageal cancer [62,63]; examining the tumour microenvironment, which is thought to be essential, e.g., for breast cancer progression and metastasis [64].

Beyond questions, the role of non-coding variants in cancer genome is significant and should be incorporated into diagnostic and treatment procedures, which in fact is being preceded by several guidelines-producing bodies including ACMG (for example [65,66,67]). WGS of cancer genome allows to characterise the whole profile of genetic variants and assign them to a proper cancer signature or specific feature. Each of the more than a hundred signatures identified up to date across human cancers indicates a specific mechanism of cancer development [68]. Most of the signatures can be associated with a defective DNA maintenance process and a precisely pinpointed disrupted pathway, which brings us to the point where specific treatment may be administered on the basis of this information, such as PARP inhibitors.

PARP inhibitors (poly-(ADP-ribose)-polymerase inhibitors) are ground-breaking agents, effective in treating several cancer types including breast, ovarian, prostate and even pancreatic cancer [69,70,71,72,73,74]. Multiple randomised clinical trials have demonstrated their efficacy and the PARPi drug family constantly expands, comprising such agents as Olaparib, Niraparib, Rucaparib and Talazoparib, with many more under clinical trials around the world [75,76,77]. However, it remains worrisome that only a subset of cancer patients treated with PARPi actually benefit from the therapy [78,79].

The biomarker currently used for PARPi administration is far from being perfect: the *BRCA1* and *BRCA2* gene mutations [80,81]. Even though they are an excellent indicator of Homologous Recombination Deficiency (HRD), they are not the only hallmarks of HRD disruption [82]. However, clinical trials have clearly demonstrated that patients without *BRCA1* nor *BRCA2* mutations can also benefit from PARPi therapy [83]. For example, the PRIMA trial (PRIMA/ENGOT-OV26/GOG-3012 trial results presented at the European Society for Medical Oncology (ESMO) Congress in 2019) showed that assessing HRD status with the aid of computer algorithms may allow more cancer patients with no *BRCA1* and *BRCA2* mutations to undergo a successful PARPi treatment [83]. Thus, many more patients without *BRCA1* and *BRCA2* do respond to PARP inhibitors and therefore may benefit from the treatment.

In fact, the most advanced clinical application originating from cancer signatures, not only mutated genes, relates to Homologous Recombination Deficiency and PARPi [84,85]. WGS is being used in a couple of commercially available cancer diagnostics; for example, Illumina Comprehensive Genomic Profiling considers Tumour Mutation Burden (TMB) or Microsatellite Instability (MSI) or MyChoiceCDx created by Myriad Genetics Inc. The diagnosis and treatment based on advanced machine learning algorithms, such as HRDetect or myChoice already show promising results: several clinical trials of the drugs based on PARPi (such as Niraparib, most recently) were effective in reducing the risk of ovarian cancer progression by 38% [85,86,87]. AI-based computer algorithms are created to screen WGS data for rare and common variants potentially significant in pharmacogenomics, leading to new applications of the drugs already existing in the market, but also identification of novel regulatory variants located in non-coding parts of the genome and their function, patient stratification and, in some cases, even the mechanistic prediction of drug targets, response and their interactions [88,89]. Some cancer databases are designed to find patient target genes and potential treating molecules [90]. Although datasets contain various omics datasets, such as mRNA and epigenomics, WGS data are still the core of such databases. As a result, a hit containing a list of potential drugs targeting a particular genetic sequence is returned.

Regional databases also play an important role in cancer research. Numerous studies are focused on specific populations, such as 237 patients from a reported population-based south Swedish triple-negative breast cancer cohort profiled by RNA sequencing and whole-genome sequencing included in “Molecular analyses of triple-negative breast cancer in the young and elderly” or a population-based Estonian biobank (over 150,000) and breast cancer-affected cases from Latvia chosen to assess the spectrum and frequency of CHEK2 variants in the breast cancer-affected and general population in the Baltic states region [91,92].

## 4. Genomic Databases in Infectious Diseases

The same is true for many other human threats including infectious diseases. It has been long known that not only can we track pathogens’ routes of transmission or evolutionary development, as it has been done for MRSA strains [93,94] or cholera outbreaks in Haiti [95,96], but also genomic regions in human DNA connected with susceptibility or resistance to a certain pathogen, such as norovirus infections [97,98,99]. More recently, this phenomenon was beautifully depicted by the global cooperation established at the very early days of the COVID-19 pandemic, namely the COVID-19 Host Genetics Initiative (HGI) and the COVID Human Genetic Effort (HGE). These global initiatives aimed at understanding the disease enabled worldwide genomic sample collection, used further by us and others, and resulted in enormous datasets suitable for AI- and ML-based algorithms (exemplified by the HGI and HGE consortia findings described in [100,101]. Such great databases provide evidence that, as a scientific community worldwide, we are already very good at collecting data, but the time has come to share these datasets more eagerly. Especially in case of the genomic datasets, it may not be feasible nor technically doable for a single team to analyse and interpret properly whole genome sequences of such a huge and expanding collection.

It is worth emphasizing that all the genomic data collected during the COVID-19 pandemic can be used not only in the infectious context. Our project “Search for Genomic Markers Predicting the Severity of the Response to COVID-19” may be taken as an example. Between April 2020 and April 2021, we collected samples from 1222 Poles to study their genetic susceptibility to COVID-19 infections. We analysed the whole genomes to identify and genotype a wide spectrum of genomic variation, such as small and structural variants, runs of homozygosity, mitochondrial haplogroups and de novo variants. This study is the biggest whole-genome screening of the Slavic and Central Europe populations done to date. The allele frequencies, calculated for 1076 unrelated individuals, were released as a publicly available resource, the Thousand Polish Genomes database. The Polish population, highly homogenous and sedentary by its nature, is unique and can serve as a genetic reference for the Slavic nations that account for over 4.5% of world inhabitants. The Thousand Polish Genomes database contributes to the worldwide genomic resources accessible to researchers and clinicians. It lays the foundation for further studies in the population history and epidemiology of diseases caused by mutations in the autosomal-recessive genes, as well as creates opportunities for tailoring NGS-based genetic screening tests and guidelines for clinical geneticists in Poland [48].

Genomic databases in infectious diseases can play multiple roles not only in relation to COVID [102]. They may help in identification of resistance biomarkers and treatment targets. This seems to be particularly crucial in Africa, especially for the detection and surveillance of malaria, HIV and drug-resistant tuberculosis [103].

As communicable diseases quite often have a localised character, creating small, local databases might be particularly useful in their case. During the 2019–2020 dengue fever epidemic in the Dominican Republic, a study on 488 children with a confirmed disease was conducted to find the genetic factors of its severity in this group [104]. On the African continent, there is a need to investigate tropical arboviruses with described zoonotic potential. The whole-genome sequencing using novel technological approaches allows a better understanding of their genetic diversity and distribution that may help to reduce the threat they pose to human and animal health [105]. The large international databases are also frequently used in this domain. In Asia, 10 *Pasteurella canis* and 16 *Pasteurella multocida* whole-genome sequences from National Center of Biotechnology database were selected to perform a comparative analysis of virulence factors (VFs) between two species that both cause zoonotic infections [106]. The collections such as the Comprehensive Antibiotic Resistance Database (CARD) or the Virulence Factor Database (VFDB) are used to identify the genes responsible for drug-resistance or virulence and characterise local pathogens. It was done recently in the case of multidrug-resistant Staphylococcus hominis isolated in Malaysia [107].

## 5. Genomic Databases in Rare Diseases

Rare diseases were one of the areas that profited from the WGS technology at first. Moreover, in terms of standardisation and guidelines, WGS in rare diseases is well established [108]. One of the most known and pioneering initiative is 100,000 Human Genomes, a project targeted at sequencing NHS patients affected with rare diseases [109]. The preliminary results gave a diagnostic yield of 35% for likely monogenic disorders and 11% for likely complex disorders [109]. In the US, Centers for Mendelian Genomics are pioneering institutions that use WGS in rare disease cohorts [110]. In Canada a centralized WGS database for rare diseases has been introduced to facilitate cooperation and new gene discovery [111]. On the European level several EU-founded projects, such as Solve-RD and ERN, implemented WGS as part of their workups [112]. In the recent years also regional initiatives, such as the Brasilian Rare Genomes Programme [113] and the Initiative on Rare and Undiagnosed Diseases in Japan [114], have successfully been implemented.

## 6. Genomic Databases as a Fuel for AI-Driven Algorithms

Furthermore, it is worth noting that AI-based methods already are and will remain an integral component of every modern WGS-based procedure. We can develop AI-driven algorithms to extract crucial information from the patient’s genome and include them into prediction or prognostic tests even without full understanding of the region itself. This is a major breakthrough seriously challenging our perception of the scientific method. So far, the sequence of diagnostic and therapeutic actions was preceded by deep understanding of the target itself, its structure and function, such as the gene. Using various techniques, we have been studying genes and their role in model organisms for years before transferring this knowledge into human beings. However, now AI-driven algorithms may pinpoint genomic regions or clusters of unknown function, yet crucial for the early prediction, advanced diagnosis, or effective treatment. Although Al technologies have raised an initial enthusiasm, there are also critical voices. For example, in a review article on AI methods in diagnosing COVID 19, the authors found methodological flaws and biases leading to an optimistic performance. The authors advise standardisation of methodology on several levels [115].

Although several multiple large-scale whole genome sequencing projects have been launched globally, and the results obtained so far are important both scientifically and clinically, the clinical implementation of these data is for the most part lagging behind. Most of the projects are focused on rare diseases and clinical WGS was primarily used as a rare disease’s diagnostics. Together with whole-exome sequencing (WES), WGS has been introduced into diagnostic procedures of many clinical centres, such as Genomic Medicine Centre Karolinska-Rare Diseases in Stockholm or Genomics England centres across the UK. There are many more similar institutions and programs worldwide focusing on clinical WGS in rare disease diagnostics, such as several National Institutes of Health grant programs in the US, the Clinical Sequencing Exploratory Research Consortium, the Centres for Mendelian Genomics, and the Undiagnosed Diseases Program and Network [110,116]. The advantage of WGS is estimated as the 7.5–30% increase in diagnostic yield. However, WGS seems to be promising because of the diversity of variants detected, difficult to find using other available methods, including CNVs, balanced structural variants, short tandem repeats and runs of homozygosity [117].

Another challenge is the identification of common and rare disease genetic variants in genome-wide association studies. WGS together with dedicated AI-driven algorithms was shown to increase the mapping precision for rare and low frequency variants. More and more, WGS is being performed in a variety of different populations, supporting the notion that WGS of related cohorts improves the power to identify genetic associations [43,118,119,120,121].

The most significant drawback when using AI-harnessing algorithms, apart from the costly IT infrastructure, remains the huge dataset necessary for the proper learning process. Although several mathematical models can overcome this problem, at least to some extent, the reliability of the results and their clinical implementation should be strongly considered and properly validated. Perhaps the next generation of AI-based genomic-analysing tools are required and thus, should emerge from interdisciplinary close cooperation.

Finally, another objection—the time required to complete the process—is no longer valid. Although our project was performed using a “traditional” short-reads approach and Illumina pipeline for the WGS data, there are already other methods which might be better, especially given the narrow time constraints in the case of some rapidly progressing diseases. One of the most amazing examples of interdisciplinary cooperation on the ground of AI-driven tools implementation in clinical genomics was a recent world record in whole-genome sequencing speed, counted from the moment of sample arrival till results delivery. A Stanford University research team led by Dr Euan Ashley, in collaboration with such technological giants as Nvidia, Oxford Nanopore Technologies, Google, as well as the medical world-famous Baylor College of Medicine and the University of California, managed to complete the process in just five hours and two minutes [122].

Although there is a lot of hope in the AI-based methods, too early translation to the clinics may lead to wrong conclusions and failures in treatment. All algorithms are often trained on a single centre’s data and may be biased. For example, surgical skin markings confused a deep learning algorithm for melanoma detection in which it classified benign nevi as malignant [123] As another example, an AI system recommended “unsafe and incorrect” cancer treatments [124]. Similarly, a sepsis prediction algorithm implemented in a widely used EHR system performed poorly in practice. [125] A special care should be given as most of the projects are targeting clinical research [126].

## 7. Conclusions

Genomic databases are not only collecting data but may also contribute to better diagnostics and therapies. Genomic databases play a special role in infectious diseases, as well as rare or heritable diseases. As a medical community, we should make the most of what we have already achieved in genomics to effectively treat cancer patients. Offering the most advanced diagnostic methods or early detection tests today, we should continuously participate in building more accurate prediction models especially for early detection or targeted therapies.

## Figures and Tables

**Figure 1 ijms-24-03031-f001:**
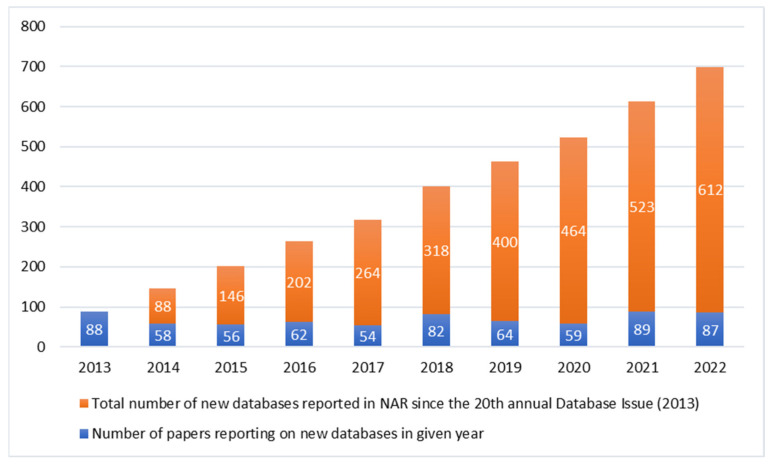
New databases reported in *Nucleic Acids Research* database issue in the last 10 years.

**Figure 2 ijms-24-03031-f002:**
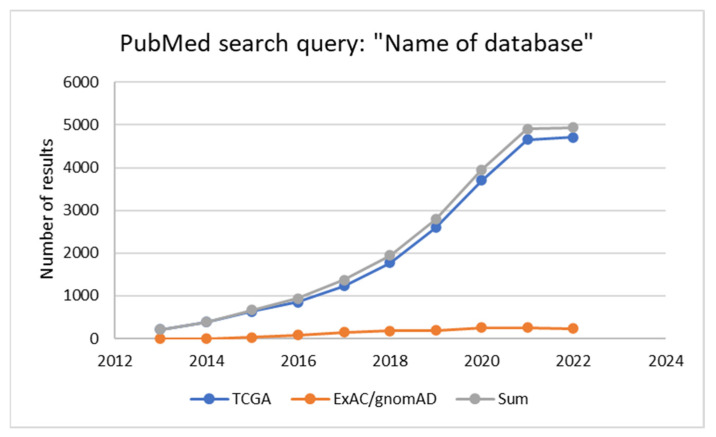
Number of results of the PubMed search query “Name of database” for two databases described in the review.

**Figure 3 ijms-24-03031-f003:**
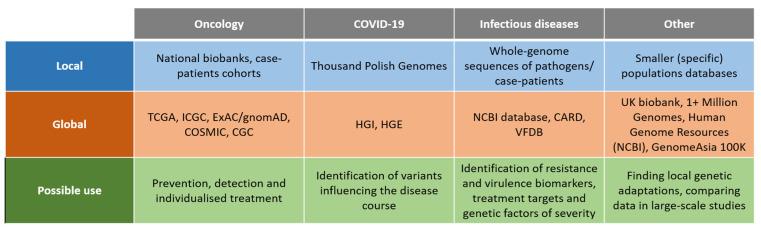
Sources of genomic data (existing databases) described in the review. Note that not all of the databases mentioned above contain WGS-derived data; some of them were created on the basis of other techniques.

## Data Availability

Not applicable.

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
