# Peer review of "WGS Data Collections: How Do Genomic Databases Transform Medicine?"

_ijms, 2023, doi:10.3390/ijms24033031_

Round 1
Reviewer 1 Report
The authors present genomic databases and their medical applications. They compiled various databases and the information would be very useful for many researchers. However, there are several issues to be improved.
1. It would be helpful to show information of the databases using a table. Although they show the information at Figure 3, it needs to more detail description for each database by a table.
2. They focused on two kinds of disease, cancers and infectious diseases. Why did not they present other types of databases?
3. The authors present 'Genomic databases as a fuel for AI-driven algorithms' at Page 8. However, the characteristics of this section appears to be distinct from other sections. Moreover, the datasets of other sections might be applicable to the AI-based algorithms.
Author Response
Dear Reviewer,
Many thanks for the time and effort which you and the reviewers took to review the above-referenced manuscript. We greatly appreciate the helpful suggestions for improving our manuscript and have adapted the paper, accordingly, as also detailed in your point-by-point response. As a result of this revision, we believe that our manuscript has been greatly improved – including added information about the databases, only mentioned briefly in a previous version. Also, abbreviations from the Figure 3 have been explained, as well as databases info added in a text. New paragraph about rare diseases database has been added and, in several places, existing text has been enhanced with new information or sources. Above all, we totally agree that the datasets of other sections might be applicable to the AI-based algorithms, so we tried to highlight this common point in the final paragraphs. Therefore, we would be grateful if you consider re-evaluating our manuscript in its revised form for potential publication in the journal.
We ensure that any revisions made in response to reviewer comments regarding patient-level data are compliant with privacy and data protection laws. All authors have read the revised manuscript and agreed to publish it in the present form.
Yours sincerely,
On behalf of all authors,
Dr. Zbigniew Król, MD, PhD

Reviewer 2 Report
Authors reviewed the benefits of whole-genome sequencing (WGS) to identify non-coding genomic regions for advancing medicine, for example, effective PARP inhibitors in oncology and susceptibility to COVID-19 or severity in infectious diseases. Although existing databases are useful for data mining, de novo sequencings for long non-coding genomic regions, such as repetitive sequences in centromeres and acrocentric chromosomes that make up over 6% of human genome, are also essential. It is also important to list the technologies used for obtaining the genomic data in Figure 3, as some databases contain data from genome hybridization, not whole genome sequencing.
Author Response
Dear Reviewer,
Many thanks for the time and effort which you and the reviewers took to review the above-referenced manuscript. We greatly appreciate the helpful suggestions for improving our manuscript and have adapted the paper, accordingly, as also detailed in your point-by-point response. As a result of this revision, we believe that our manuscript has been greatly improved – including added information about the databases, only mentioned briefly in a previous version. We absolutely agree that the technologies used for obtaining the genomic data mentioned in our text, especially in Figure 3, are not solely from whole genome sequencing (WGS), but also other techniques, such as genome hybridization. Our initial point was to focus on the datasets as such, in order to familiarize readers with further possibilities, maybe even encourage to use the existing databases – in our opinion amazing source of information and new knowledge. Therefore, we would be grateful if you consider re-evaluating our manuscript in its revised form for potential publication in the journal. We ensure that any revisions made in response to reviewer comments regarding patient-level data are compliant with privacy and data protection laws. All authors have read the revised manuscript and agreed to publish it in the present form.
Yours sincerely,
On behalf of all authors,
Dr. Zbigniew Król, MD, PhD

Reviewer 3 Report
I think it is an important manuscriot that the authors presents the examples of the existing and currently being developed genomic databases and their possible use. I would like to make three comments.
#1 How about a few more references in the first paragraphs of the introduction? (For example, line 4 to 6)
#2 In the second paragraph of section "Genomic databases as a fuel for AI-driven algorithms", I would appreciate it if you could provide the references on which you based your opinion.
#3 I believe the authors have obtained and reviewed information from various papers about the special role of genome databases in infectious, rare, and genetic diseases. I would like to see a more specific description of the emerging role of genome databases in the treatment of cancer patients, if possible.
Author Response
Dear Reviewer,
Many thanks for the time and effort which you and the reviewers took to review the above-referenced manuscript. We greatly appreciate the helpful suggestions for improving our manuscript and have adapted the paper, accordingly, as also detailed in your point-by-point response. As a result of this revision, we believe that our manuscript has been greatly improved – including added information about the databases, only mentioned briefly in a previous version. Also, abbreviations from the Figure 3 have been explained, as well as databases info added in a text. New paragraph about rare diseases database has been added and, in several places, existing text has been enhanced with new information or sources, especially on rare diseases and oncology. Moreover, it is worth mentioning that our initial point was to focus on the datasets as such, in order to familiarize readers with further possibilities, maybe even encourage to use the existing databases – in our opinion amazing source of information and new knowledge. Therefore, we would be grateful if you consider re-evaluating our manuscript in its revised form for potential publication in the journal. We ensure that any revisions made in response to reviewer comments regarding patient-level data are compliant with privacy and data protection laws. All authors have read the revised manuscript and agreed to publish it in the present form.
Yours sincerely,
On behalf of all authors,
Dr. Zbigniew Król, MD, PhD

Round 2
Reviewer 1 Report
The authors addressed the previous concerns.